# Eudermic Properties and Chemical–Physical Characterization of Honeys of Different Botanical Origin

**DOI:** 10.3390/nu16213647

**Published:** 2024-10-26

**Authors:** Elisabetta Miraldi, Giorgio Cappellucci, Cecilia Del Casino, Emanuele Giordano, Massimo Guarnieri, Massimo Nepi, Marco Biagi, Giulia Baini

**Affiliations:** 1Department of Physics, Earth and Environmental Sciences, University of Siena, 53100 Siena, Italy; giorgi.cappellucci@unisi.it (G.C.); giulia.baini2@unisi.it (G.B.); 2Department of Life Sciences, University of Siena, 53100 Siena, Italy; cecilia.delcasino@unisi.it (C.D.C.); emanuele.giordano2@unisi.it (E.G.); massimo.guarnieri@unisi.it (M.G.); massimo.nepi@unisi.it (M.N.); 3Department of Food and Drug, University of Parma, 43121 Parma, Italy; marco.biagi@unipr.it

**Keywords:** honey, chemical–physical analysis, antioxidant, skin protection

## Abstract

**Background:** Honey is a natural product that, thanks to its composition, particularly the high sugar content, is highly appreciated as an energy nourishment. In addition to sugars, it contains many other substances (carbohydrates, free amino acids, enzymatic proteins, organic acids, polyphenols) from which the therapeutic properties of honey arise: hydrating and osmotic activity, antimicrobial action, and antioxidant and anti-inflammatory power. **Objectives:** The present work aims to deepen our knowledge/understanding of the activity of skin protection exerted by honey, as a synergic result of its multiple therapeutic effects. Moreover, this study wants to find possible correlations between biological properties and the chemical–physical traits of honey. **Methods:** To carry out this research, five varieties of citrus honey, one of acacia honey, one of chestnut honey, and one of multifloral honey were used. The honeys were first characterized by chemical–physical analysis and then were subjected to qualitative melissopalynological analysis. Tests were also carried out to evaluate both their antioxidant power and the effect on collagenase, an enzyme involved in the degradation of collagen present in the extracellular matrix and, therefore, in the processes of skin aging. Finally, honey samples were then used in in vitro experiments to assess their action in stimulating cell viability and proliferation on human keratinocytes. **Results:** Chemical–physical analysis demonstrated a good water content (about 17%), an important sugar content (with the monosaccharides glucose and fructose being the most represented in all the honey samples), various amino acids (with proline remarkably being the highest in all honeys), and a high concentration of polyphenols and total flavonoids (the maximum in chestnut honey, 762 mg/kg and 514 mg/kg, respectively). **Conclusions** The results obtained in this work confirm the ethnopharmacological use of honey in wound care, bring new scientific knowledge on the use of honey in dermatology, and highlight two fields of excellence, particularly incitrus and chestnut honey.

## 1. Introduction

Honey, as stated in Legislative Decree No. 179 of 21 May 2004 [1], is ‘the natural sweet substance that bees (*Apis mellifera* L.) produce from the nectar of plants or from secretions from living parts of plants or from substances secreted by sucking insects on living parts of plants that they forage, transform, combine with their own specific substances, deposit, dehydrate, store and allow to mature in the honeycombs of the hive’ [2].

Attracted by the bright colors and smells of the flowers, the foraging bees use a special mouth structure, the ligule, to collect the nectar and hold it inside an esophageal appendage, the bursa melaria, before regurgitating it back into the hive, where it will go on to the next stages of honey production thanks to the activity of the worker and fan bees. The honey, which is not yet ripe, is laid in thin layers and dehydrated by the fan bees, which, by swirling their wings, generate air flows that cause water to evaporate from the honey until an ideal aqueous concentration for storage is reached. From the foraging stage, the conversion of the sucrose present in the nectar into monosaccharides, glucose and fructose, begins through the catalytic activity of the invertase enzyme with which bees are equipped. When the honey is mature, it is finally stored and sealed in the comb cells [2].

Honey is a highly energetic food due to its high concentration of monosaccharides, particularly glucose and fructose, which are rapidly assimilated without requiring a digestive process. The presence of disaccharides such as maltose and sucrose, as well astrisaccharides like melezitose, contributes to the sugar composition of honey. The water concentration in honey, influenced by various factors such as botanical origin, production season, and atmospheric conditions, is crucial for its preservation and quality [3].

Honey contains a small percentage (0.2–2%) of nitrogen compounds, including free amino acids and enzymatic proteins such as glucose oxidase and invertase. The presence of amylase contributes to the breakdown of starch into glucose. Honey also contains minerals, primarily potassium, absorbed by plants and present in nectar. Both inorganic and organic acidic compounds, including gluconic acid, are present [4].

Traces of vitamins, mainly vitamin C and B-group vitamins, are found in honey due to pollen grains. Minor constituents such as aldehydes, ketones, esters, ethers, and phenols influence the organoleptic properties of honey, affecting its color and aroma. Phenolic compounds, though present in small quantities, play a significant role in the beneficial properties of honey [5].

Honeyhas proven healing properties supported by contemporary scientific evidence. Acknowledging its dual nature as both nourishment and medicine, scientific research has validated honey’s efficacy in accelerating wound healing and re-epithelialization [6,7,8].

Honey’s therapeutic effects result from a synergistic combination of its antibacterial activity [9,10,11], osmotic and moisturizing action, and high viscosity [12]. These properties collectively create a protective barrier that hinders pathogen penetration and fosters an optimal environment for rapid healing.

Described as eudermic, honey combines antimicrobial, antioxidant, and anti-inflammatory properties with emollient, soothing, and moisturizing actions, leading to improved skin turgidity, elasticity, and tone.

In the three most usual wound categories (human burns, ulcers, and other wounds), honey seems to be a dressing with wound-healing stimulating properties, as reported by Vandamme et al. (2013) [7]. In burns, there is also evidence for its antibacterial capacity.

Honey’s therapeutic prowess is rooted in its chemical and physical characteristics. Its high sugar content, hygroscopicity, acidic pH, and rich concentration of antioxidants, such as polyphenols and flavonoids, contribute to its healing potential [13,14]. Key amino acids present in honey, particularly proline, play a crucial role in the re-epithelialization phase, accounting for a significant portion of collagen composition [15].Experiments on human keratinocytes demonstrate honey’s ability to induce the expression of extracellular matrix metalloprotease (MMP-9), an enzyme involved in collagen degradation during tissue remodeling in wound repair [16,17,18].

This study has two different aims: first, the chemical characterization of different honey types, specifically five citrus varieties (*Citrus* spp. L.) produced from nectar collected from the flowers of plants of the ‘*Citrus*’ genus (to which lemon, orange, grapefruit, mandarin, citron, and bergamot belong), one variety of acacia honey (*Robinia pseudoacacia* L.), one of chestnut honey (*Castanea sativa* Mill.), and, finally, one of multifloral honey.Chemical characterization is very important because the phytocomplex strongly influences biological activity.

Secondly, we determined the eudermicactivity of the tested honeys, in order to assess their natural efficacy in exerting skin protection activity.

This comparative study on Italian honeys with different chemical–physical characteristics and botanical origins adds new information about this bee product that can represent an important resource for skin care; the obtained results give positive experimental feedback to what was suggested in literature regarding the use of honey in dermatology.

## 2. Materialsand Methods

### 2.1. Samples

Five different citrus (*Citrus* sp.) honeys were from lowland citrus-growing areas in the province of Chieti (Italy), while one acacia (*Robinia pseudacacia* L.) honey, one chestnut (*Castanea sativa* Mill.) honey, and a multifloral honey were from Montalcino (Siena, Italy). All the honeys were produced in 2018.

The samples were stored at +5–7 °C and in the dark, which are optimal storage conditions to preserve the chemical and physical properties of the honey.All the chemical analyses were performed in sextuplicate, and data are expressed asthemean ± standard deviation.

### 2.2. Water Content

Water content measurement was performed using a portable refractometer, theATAGO HHR-2N (ATAGO, Milano, Italy), calibrated at a temperature of 20 °C.

A small quantity of the honey sample was spread on the prism provided, and after closing the glass lid and turning the refractometer toward the light, it was possible to read the value of the water content (expressed as percentage by weight, g of water per 100 g of honey). For measurements made at temperatures other than the calibration temperature, the values returned by the refractometer were corrected by the addition of a factor automatically determined by the instrument itself.

### 2.3. Determination of the Sugar Profile

The sugar profile was determined by a Waters LC-Module 1 HPLC system coupled to a Waters RI 2410 refractive index detector (Waters Corporation, Milford, MA, USA). Honey samples were diluted 1:20 with Milli-Q water. Separation was carried out in isocratic elution, with a flow of 0.5 mL/min, using water as the mobile phase and a Waters Sugar Pack column (300 × 6.5 mm, particle size 10 µ), thermostatically controlled at 90 °C. Analysis was carried out on 20 µL of sample, inoculated by injection. Retention times in minutes of analytes were as follows: pectins, 4.9; sucrose, 8; glucose, 10; fructose, 11.4; melezitose, 6.9. The concentrations of each sugar were calculated by comparing the area under the chromatogram peaks with standards of sucrose, glucose, fructose, pectin (galacturonic acid), and melezitose (Sigma-Aldrich, St. Louis, MO, USA, purity ≥ 98%), using Clarity CSW-32 software (https://ppm.itdesign.de/)(Tubinga, Germany).

### 2.4. Determination of Free Amino Acids

The determination of free amino acids was performed by the Waters LC-Module 1 HPLC system coupled with a Waters 470 Scanning Fluorescence detector, using a method taken from Ma et al. (2015) [19]. Because the amino acids are not themselves visible by fluorimetry, it was necessary to derivatize them by employing 6-amino quinolyl-N-hydroxysuccinimidyl carbamate (AQC), which reacts with primary and secondary amino acids, converting them into stable fluorescent derivatives.

An AccQ-Fluor reagent kit (Waters Corp., Milford, MA, USA) was used for derivatization. External standards were used as a reference for the determination. Hydrolyzed amino acid standard mixtures including 2.5 mM aspartic acid (Asp), glutamic acid (Glu), serine (Ser), glycine (Gly), threonine (Thr), alanine (Ala), histidine (His), proline (Pro), arginine (Arg), cysteine (Cys) (1.25 mM), valine (Val), tyrosine (Tyr), methionine (Met), isoleucine (Ile), leucine (Leu), phenylalanine (Phe), and lysine (Lys) were obtained from Waters Corp. The individual amino acids ornithine (ORN), ß-alanine (BALA), A-aminobutyric acid (AABA), ß-aminobutyric acid (BABA), γ-aminobutyric acid (GABA), and taurine (TAU) were supplied by Sigma-Aldrich. Sodium acetate (NaAc) and phosphoric acid were obtained from Panreac Quimica. Acetonitrile (ACN) and triethylamine (TEA) were purchased from Carlo Erba s.p.a. Milli-Q grade water was used for the analyses. The separation was carried out using a SupelcoAscentis C18 column (250 mm×4.6 mm × 5 µm). Each honey sample was diluted 1:20 with Milli-Q water. Next, 70 µL of this solution was added to10 µL of AQC reagent and 20 µL of AccQ.Tag kit buffer. The solution was vortexed and then transferred to an oven at 55 °C for 15 min and then run for the separation of the amino acids. Details of the chromatography conditions are described in Appendix A). The purity of all the reagents wasHPLC grade (≥98%).

### 2.5. Determination of Total Polyphenols

The total polyphenol content (TPP) of the honeys was evaluated by means of the Folin–Ciocalteau (FC) colorimetric method as described in Finetti et al., 2020 [20]: briefly, 100 µL of honey (100 mg/mL)was diluted to 3 mLwith ddH_2_O; 500 µL of FC reagent diluted 10-fold in water (Sigma-Aldrich, Milan, Italy) wasadded; the mixture was shook;and, finally, 1000 µLof a 30% *w*/*v* sodium carbonate water solution wasadded. After incubation for 1 h in the dark at room temperature, the absorbance of the samples was read at 750 nm using ddH_2_O. Gallic acid (Sigma-Aldrich) was used as the reference standard. A calibration curve was created using different solutions of gallic acid (5000 to 78 mg/L), testing 10 µLof each of them.

### 2.6. Total Flavonoid Content of Extracts

The total flavonoid content (TF) of different extracts was evaluated by means of spectrophotometric quantification and expressed as quercetin [21]. Briefly, samples (100 mg/mL) werediluted 10-fold in distilled water, and the absorbance was recorded at 353 nm, which is the level of maximum absorbance of quercetin (Sigma-Aldrich, Milan, Italy), used as a reference standard (calibration curve: absorbance read in the range 5–0.078 mg/L).

### 2.7. Qualitative Melissopalinological Characterization

A total of 10 g of honey (15–20 g in the case of pollen-poor honeys) was weighed into a conical-bottomed test tube, to which 20 mLof distilled water (temperature not exceeding 40 °C) was added. The tube was subjected to centrifugation for 15 min at 2500 rpm. After this was finished, the supernatant was removed and the sediment was resuspended with 10 mLof distilled water. Then, further centrifugation was carried out for 10 min at 2500 rpm. The obtained supernatant was removed, and the sedimentwascollected and spread evenly on a glass slide. This was covered and sealed with a coverslip on which a drop of glyceratedgelatin was placed. Finally, the microscopic observation (400× or 1000× magnification), recognition, and counting of pollen grains were carried out. Recognition was performedby comparison with melissopalinologicalatlases [22,23] and with reference preparations. Counts were performed on two slides prepared from the same honey independently to obtain greater reliability of the figure calculated as the average of the two values.

### 2.8. Antioxidant Activity

The radical scavenging activity of the honey samples was measured by means of the DPPH assay, as previously reported [24]. Briefly, 10 µL of different concentrations (2–200 mg/mL) of the samples were added to 190 µL of freshly prepared methanolic DPPH solution (0.1 mM) and incubated for 30 min at rt in the dark with shaking. Then, absorbance was recorded at 517 nm using a Victor^®^Nivo™ plate reader (PerkinElmer, Waltham, MA, USA). ddH_2_O was used as the blank control. The antiradical activity of the samples was calculated according to the following formula:Antiradical activity% = (Absblank − Abssample)/Absblank × 100(1)

Data were plotted using Microsoft Excel (version 2408) for each sample.

### 2.9. Collagenase Inhibition Assay

The inhibitory effect of collagenase was analyzedusing acollagenase activity colorimetric assay kit (Sigma-Aldrich, Milan, Italy, MAK293). The assay kit measures proteolytic degradation when collagenase interacts with synthetic FALGPA, a synthetic peptide that mimics collagen’s structure; this leads to a reduction in the absorption of 340 nm in the presence of collagenase inhibitors. The absorbance of the reaction mix was determined at 5 min intervals for 20 min. Collagenase activity (U/mL) and the subsequent percentual inhibition produced by the honey samples were calculated.

### 2.10. Cell Cultures

Human keratinocyte (HaCaT) cells were cultured in 75 cm^3^ flasks (Euroclone, Milan, Italy) in high-glucose DMEM (Dulbecco’s Modified Eagle Medium, 4500 mg/L) supplemented with 10% *m*/*v* FBS (Fetal Bovine Serum, Sigma-Aldrich, Milan, Italy) and 1% *m*/*v* L-glutamine (Sigma-Aldrich, Milan, Italy), incubating them at 37 °C in a humid atmosphere enriched with CO_2_ (5%).The cells were then removed from the flasks by employing trypsin-EDTA solution (Sigma-Aldrich, Milan, Italy).Cell counts were performed microscopically (Leica, Milan, Italy, 25×) using a hemocytometer.

### 2.11. Cell Viability

A cell viability assay was performed using Cell Counting Kit-8 (CCK-8) (Sigma-Aldrich, Milan, Italy) on human keratinocytes (HaCaT) treated for 24 h with extracts of the various honeys prepared at three different concentrations: 10 mg/mL, 1 mg/mL, and 0.1 mg/mL. The cells were seeded in 96-well plates at a density of 5000 cells per well in DMEM and 10% FBS and allowed to grow until they occupied about 70% of the same, and they were incubated for 24 h at a temperature of 37 °C in a humid atmosphere enriched with CO_2_(5%). After 24 h, the complete medium was aspirated to be then replaced by fresh medium with a lower percentage of FBS (3% *m*/*v*). Then, treatments were performed with the honeys at the different concentrations.CCK-8 was added for each well at a ratio of 1:10 to the total volume of the sample well, and after two hours, spectrophotometric readings were taken with a Victor^®^Nivo™ plate reader (PerkinElmer, Waltham, MA, USA), at a wavelength of 450 nm.

### 2.12. Proliferation Assay

The same test was repeated by also treating the cells with methylprednisolone 1 mg/mL (Euroclone, Milan, Italy), used as a cell viability inhibitor. Methylprednisolone was used alone or in cotreatment with the honeys. After 24 h, itwas aspirated and replaced by fresh medium. CCK-8 was added at a ratio of 1:10 to the total volume of the well and, after two hours, spectrophotometric readings were taken with a Victor plate reader at a wavelength of 450 nm.

### 2.13. Statistical Data

Principal component analysis (PCA) was performed using the open-source software R (version 4.0.3, base library), operating on a dataset consisting of the results of amino acid and carbohydrate determination of all analyzed honey samples. The data matrix was standardized (as the observations were expressed in different units), applying the z-standardization technique, before extracting the principal components, whose number was chosen based on the methodology proposed by Jolliffe (1972) [25].The choice of PCA technique to obtain clustering of observations (instead of cluster analysis) was driven by the desire to seek a chemical–physical interpretation of the classification, through the dimensionality reduction of data, summarized in the new derived variables. Furthermore, PCA also allows for visually depicting relationships among latent variables, which is essential for the objectives of this study, aimed at investigating causes and differences in the eudermic properties of different types of honey.

The statistical difference of the test results between the different samples was determined by analysis of variance (ANOVA) followed byapost hoc Tukey test. Statistically significant values were those with *p* < 0.05 against the considered reference. Graphs and calculations were performed using Microsoft Excel and GraphPad Prism (version 10.3.1). 

## 3. Result and Discussion

### 3.1. Determination of Water Content

The water content of all honeys is less than or equal to 18% (Table 1), a condition that would prevent the occurrence of fermentation processes that would compromise the quality of the product. So, all samples have an optimal moisture content to ensure good honey preservation.

Among the essential benefits offered by honey in the treatment of skin wounds, it assures a moist environment because it has the property of being nonadherent. Then, the integrity of the skin surface is maintained and is responsible for providing a barrier for the bacteria, which can prevent cross-infection and contamination with microbes. Numerous investigations have indicated that its antibacterial activity eradicates a significant variety of microbes, which speeds up the process of wound healing [26,27]

### 3.2. Determinationof Sugars

The carbohydrates most commonly represented in all the honey samples are the two monosaccharides glucose and fructose, with the concentration of fructose slightly higher than that of glucose (Table 2).

Traces of the trisaccharide melezitose and pectinsareare also present in rather similar concentrations in citrus honeys, but they arehigher in acacia, chestnut, and multifloral honeys. Higher sucrose concentrations are recorded in multifloral and chestnut honeys; however, the dominant sugars are always glucose and fructose, in agreement with data from the literature [2].

The antibacterial effect of honey can be divided into peroxide and non-peroxide components. The peroxide component is based on the activity of hydrogen peroxide. The non-peroxide components are based on phytochemicals, the acidity of the honey, and principally on high sugar content [28]. For this reason, it is very important to know the sugar composition of honey.

The presence of non-peroxide antibacterial factors in honey is proven by the persistent activity in honey products modified with catalase to eliminate the hydrogen peroxide. These antibacterial activities are also influenced by the floral source collected by honeybees, and therefore, not all honey products possess this type of property. Its antioxidant elements are very important and are responsible for the elimination of bacterial infections. Moreover, honey contributes to the production of antibodies and cellular elements implicated in the immunity system [27].

Knowing the concentration of sugars in honey samplesis importantbecause it is well known that solutions of high osmolarity, such as honey, glucose, and sugar pastes, inhibit microbial growth because the sugar molecules tie up water molecules so that bacteria have insufficient water to grow. Therefore, high osmolarity is valuable in the treatment of infections because it prevents the growth of bacteria and encourages healing. Sugar was used to enhance wound healing for several hundred patients. It has been claimed that the sugar content of honey is responsible for its antibacterial activity, which is due entirely to the osmotic effect of its high sugar content [29].

### 3.3. Determination of Free Amino Acids

In honey, there are different types of amino acids with physiological importance, such as arginine, proline, cysteine, glutamic, and aspartic acid [27,30].

The quality profile of free amino acids in the analyzed honeys includes: glutamic acid, alanine, β-alanine, arginine, asparagine, phenylalanine, glycine, isoleucine, histidine, leucine, lysine, ornithine, proline, serine, taurine, tyrosine, threonine, and valine.

Among the various amino acidswequantified, the concentration of proline is remarkablythehighestin all honeys, with a maximum value of 4866 nmol/mL recorded in multifloral honey (Table 3). This evidence is particularly significant given the role of this amino acid in the collagen biosynthesis process [31]. Proline was the main representative of free amino acids, as reported in the literature [32].

### 3.4. Determination of Total Polyphenols and Flavonoids and Antioxidant Power

The quantification of total polyphenols followed the validated and well-established method of Folin–Ciocalteau, whereas that regarding total flavonoids was performed according to [21] and, speaking specifically on bee products, according to [33,34]. We are fully aware that the method of aluminum chloride still remains the standard method for the quantification of total flavonoids in natural products, with its pros and cons, as recently reviewed by Shraim et al., 2021 [35], but we chose the method of direct reading of absorbance because of the possibility of testing different samples and different matrixes without reactions that could lead to different final chromophores related to different matrixes; however, this method was primarily chosen according to the reliability obtained in previous works and the lack of interferences in bee products. The method parametersare summarized in Table 4.

The results of the analysis for the determination of polyphenols and total flavonoids are shown in Table 5. Undoubtedly, a particular abundance of these antioxidant compounds emerges in chestnut and multifloral honey, which is in agreement with the relevant literature [36].

The non-peroxide activities of honey are based on the action of complex phenols and organic acids, which are well known as flavonoids. These antibacterial activities are also influenced by the floral source collected by honeybees, and therefore, not all honey products possess this type of property. Its antioxidant elements are very important and are responsible for the elimination of bacterial infections [27]. Moreover, honey contributes to the production of antibodies and cellular elements implicated in the immunity system [37]. Several investigations pointed out that antioxidant activities are related to total phenolic concentration [38].

The anti-inflammatory property in honey is contributed by phenolic compounds. Several studies have proven that phenolic compounds can inhibit the overproduction of inflammatory mediators such as NO [39], TNF-α [40], and PGE 2 [41]. Additionally, phenolic compounds act as free radical scavengers that can protect cells from cytotoxicity induced by proinflammatory mediators. Therefore, this will reduce the inflammation period in the wound healing stage and thus enhance the rate of healing [28].

The highest value for total polyphenols and for total flavonoids was registered in chestnut honey. In multifloral honey, a lower concentration was recorded for total polyphenols, while the highest value among the analyzed honeys was recorded for total flavonoids. Considerably lower concentrations of total polyphenols and flavonoids were found in citrus and acacia honeys.

### 3.5. Antioxidant Activity

A good antioxidant capacity was revealed for chestnut honey (Table 6), confirming the data in the literature [42]. In contrast, citrus honeys showed no antioxidant activity. An interesting result emerged for multifloral honey: despite its high polyphenol content, it did not demonstrate significant antioxidant activity. This result could have been influenced by the presence of other bioactive compounds, such as organic acids, catalase, glucose oxidase, amino acids, and proteins, which vary in concentration among different types of honey [43,44,45].

The antioxidant activities of tested honeys are important forwound healing and help in the eradication of microbial infections.Microorganisms exist inside all wounds, but the majority of them do not infect the wound, and the wound will eventuallyheal. This situation happens when the host’s immune system and the bioburden of the wound are in a state of equilibrium [38].

Therefore, the widespread development of antibiotic-resistant bacteria is a challenging problem. Therefore, current interest is focused on an alternative to antibiotics and conventional therapies, such as honey [28]. Several researchers have explored the effects of combining honey with antibiotics on antimicrobial activity in vitro and in vivo [46].

### 3.6. Qualitative Melissopalinological Characterization

Melissopalynological analyses are a key step in confirming the botanical origin of honeys, as they make it possible to identify the flowers from which bees foraged the nectar needed to produce the honey.

All analyzed honeys except, of course, the multifloral honey were ascertained to be monofloral varieties (Table 7). In chestnut honey, a percentage of 92% of chestnut pollen grains was calculated (over-represented species). In acacia honey, a considerable percentage of 25% was reached, being a hypo-represented species (15% acacia pollen is sufficient to define monofloral honey) [47]. The results for citrus honeys were very satisfactory, with percentages ranging from 33% to 56% citrus pollen grains (in this case, the pollen percentage required to consider honey unifloral is above 10%) [48].

On the other hand, the microscopic recognition of multifloral honey showed an abundance of chestnut pollen grains (51%) in this variety of multifloral honey(Figure 1). This finding can be correlated with the very similar values of total polyphenols and flavonoids that this honey shares with chestnut honey.

In addition to the pollens mentioned above, pollen grains of sulla (*Hedysarum coronarium* L.), grass, acacia, and goldenrod (*Solidago* sp.) were recognized in the honey sediments (Figure 1 and Figure 2). Amongst these, the percentage of sulla pollen found in citrus honeys 1 and 5 was rather conspicuous, where 21% and 22%, respectively, was calculated. However, in the sediment of citrus honey 3, the presence of honeydew was abundant (Figure 2a).

### 3.7. Collagenase Inhibition Assay

The collagenase inhibition assay, as well as the subsequent ones on cell cultures, was performed on five honeys, selecting two of the citrus honeys (1 and 3, chosen from the five honeys with high amino acid content and different pollen percentages), the chestnut honey, the acacia honey, and the multiflower honey.

The results of the analysis revealed a modest but detectable collagenase-inhibiting power of citrus honeys 3 (7.09%) and 1 (6.45%), followed by chestnut honey (5.71%), at a concentration of 10 mg/mL.

On the other hand, at the same concentration, no decreasing effect on collagenase functionality was observed inacacia honey and multifloral honey. At the lower tested concentrations, no honey provided efficacy data.

### 3.8. In VitroTest

The two tests on keratinocytes had different modes of execution and objectives. In the first test, cells cultured in low-serum medium, i.e., under conditions mimicking the normal condition of keratinocytes inanintact human epidermis, were treated for 24 h with the honeys at different concentrations (0.1, 1, 10 mg/mL), and the objective of the test was to assess the influence of the samples in modulating the metabolic response compared to untreated controls. In the second test, the cells were treated for 24 h with a cortisone that minimizes the metabolic activity of keratinocytes and mimics an aged epidermis condition. The objective of this test was to verify the possible counteracting effect of the cortisone by the honeys.

The results obtained in cell tests on human keratinocytes showed that the tested honeys all have a relevant influence in positively modulating the cellular metabolic response.

Under basal conditions, all five honeys showed efficacy in increasing keratinocyte cell metabolism(Figure 3). The two citrus honeys were the most effective, with a concentration-dependent effect. Citrus honey 3(CIT 3) at 10 mg/mL increased metabolism and cell viability by more than 50% compared to the control, which was highly significant. Citrus honey 1 (CIT 1) had a similar effect, only slightly smaller: +44% compared to the control. At concentrations of 1 and 0.1 mg/mL, both honeys significantly increased cell viability by approximately 20% and 15%, respectively, compared to the control, with no significant difference between the two honeys. Multifloral honey (Multi) showed very similar cell viability increase data at the three concentrations: about 15% compared to the control. Acacia honey (ACA) is the one that gave the least obvious results in this test: the changes in increased viability compared to the control were only significant at 1 mg/mL (+14% vs. control). Chestnut honey (Chest) showed a different behavior, i.e., an effect inversely correlated with the concentration used: the best efficacy figure was at 0.1 mg/mL (+14% vs. ctrl), and the worstwasat 10 mg/mL, with a total loss of efficacy. The reason for this effect of chestnut honey is to be found in its high polyphenol content (the highest among the samples tested), which exerts a known anti-inflammatory power that can in fact minimize the trophic effect characteristic of the honey phytocomplex; the effect is already reported in the literature and also widely observed in the pharmaceutical biology laboratory in which we carried out part of this experimental work [49].

After co-treatment with the methylprednisolone 1 mg/mL(Met 1 mg/mL), the cell viability of the keratinocytes was reduced by 20%, confirming the premise of the experimental model used. With the exception of CIT 3 at all concentrations and CIT 1 at the concentration of 0.1 mg/mL, all the other tested samples resulted in an improvement in cell viability compared to the cell groups stimulated with cortisone alone (Figure 4). The cell viability resulted in an increase of more than 50% for CIT 1 and multifloral honey at 10 mg/mL (with almost complete suppression of the negative effect of the cortisone) and for Chest at 10 mg/mL, compared to the cortisone treatment. The figure for chestnut honey is further confirmation of how the anti-inflammatory effect of polyphenols modulates the trophic activity of keratinocytes. Acacia honey showed a mild positive effect, but it was never significant as a change compared to the control. The effect of CIT 3 was noteworthy; in co-treatment with cortisone, it was worse than the cortisone itself. The reason for this effect is not clearly identifiable from the tests we performed but can plausibly be found in a synergistic interaction on the glucocorticoid receptor response of the keratinocytes.

Overall, broad considerations can be produced from the keratinocyte tests and application implications can be deduced. Alltestedhoneys proved effective in promoting theimprovement of keratinocyte viability and metabolism under normal conditions. However, CIT 1, Multi, and Chest have also been shown to increase cellular activity under conditions of slowed vitality.

### 3.9. Exploratory Data Analysis

The first two principal components (PC1 and PC2) cumulatively explain 67% of the variance (Appendix A).

This percentage makes it possible to project the representative sample vectors onto a two-dimensional space (PC1 and PC2), the graph of which is shown in Figure 5, retaining approximately 70% of the information.

The red arrows on the graph indicate the most significant original variables, represented as vectors. It is evident that all citrus honeys are located below or close to the bisectors of quadrants IV and II, whereas multifloral and chestnut honey are well separated in the upper right-hand corner. Considering that a high percentage of chestnut pollen was found in the multifloral honey, it is evident that PCA on the analyzed metabolites is a powerful tool for identifying the botanical origin of honey.

Interestingly, we find thatprecisely those honeys that were tested for collagenase-inhibiting actionare arranged along the dimension represented by PC2, ordered, from bottom to top, from most effective to least effective. Consequently, it is possible to interpret PC2 as a variable measuring the action of honey on collagenase. The amino acid that contributes most on this axis is tyrosine, whose influence on collagenase activity is unknown. Other significant amino acids on PC2 are leucine (LEU), proline (PRO), valine (VAL), and ornithine (ORN). The blue oval identifies honeys that have been shown to exert some antioxidant action, the first of which is chestnut honey, followed by acacia honey, multifloral honey, and citrushoney 5. Geometrically, it can be observed that the polyphenol carrier has the greatest influence on antioxidant activity, confirming what has been seen in laboratory tests. On PC1, the most important analytes are the amino acids serine (SER), arginine (ARG), lysine (LYS), the sugars sucrose and melezitose, and polyphenols.

## 4. Conclusions

This study allowed us to make the necessary considerations for the best use of the different honeys, also according to the needs of consumers.

With the aim of going beyond the healing effect, other elements for skin protection were also taken into consideration in this research work, particularly antioxidant action, a basic element for protection from solar radiation and atmospheric pollution and at the basis of skin aging, and activity on the enzyme collagenase, an important target in chrono- and photo-induced aging because it is responsible for the degradation of collagen produced by skin fibroblasts, which guarantees vigor to the dermal layer andthemaintenance of skin texture.

In conclusion, this research work brings new scientific knowledge on the use of honey in dermatology and highlights, in particular, two territorial excellences such as citrus and chestnut honey, with one chemically characterized by a rich amino acid profile, which plays a fundamental role in maintaining skin homeostasis [50], and the other by an abundant concentration of polyphenols, which intervene by modulating cellular aging processes [51].

## Figures and Tables

**Figure 1 nutrients-16-03647-f001:**
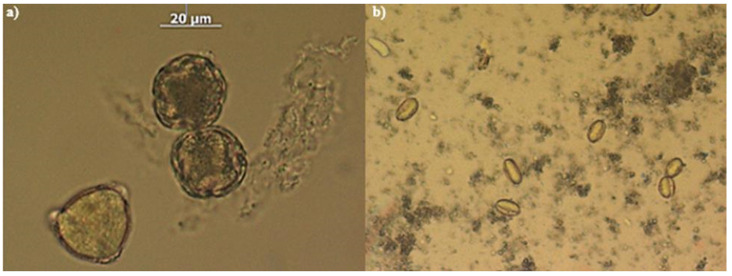
Photographs of some pollens recognized in the analyzed honeys. From left: (**a**) pollen from *Citrus* spp.; (**b**) chestnut pollen.

**Figure 2 nutrients-16-03647-f002:**
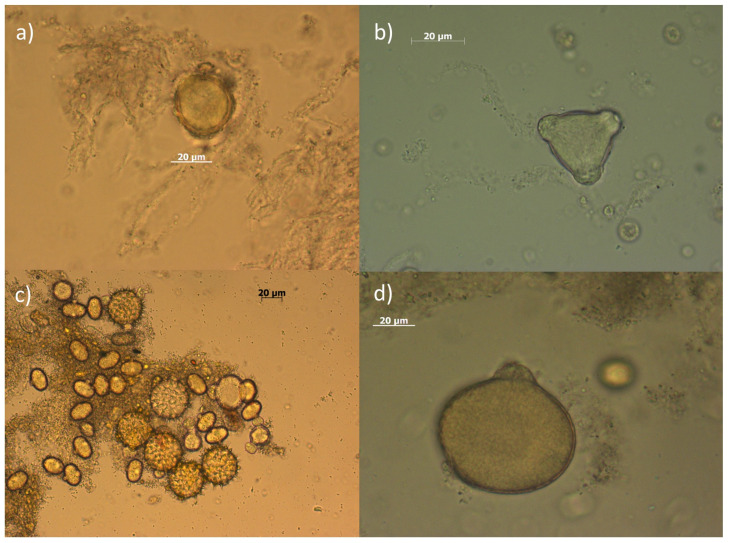
Photographs of some pollens recognized in the analyzed honeys. Top from left: (**a**) Traces of honeydew and *Citrus* spp. pollen; (**b**) Acacia (*Robinia pseudacacia*) pollen; (**c**) Pollen of sulla (*Hedysarumcoronarium*) mixed with sunflower (*Helianthus annuus*) pollen (larger grains); (**d**) Grass pollen grain.

**Figure 3 nutrients-16-03647-f003:**
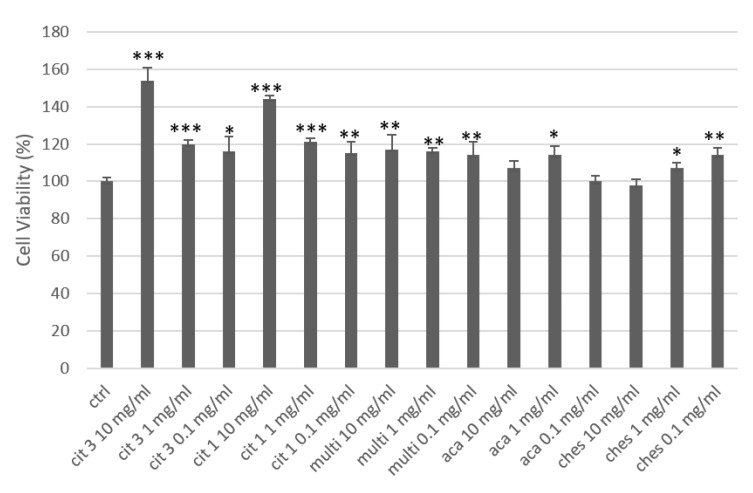
Cell viability of human keratinocytes treated with honey. * = *p* < 0.05 vs.ctrl; ** = *p* < 0.01 vs. ctrl; *** = *p* < 0.001 vs.ctrl (*t*-test).

**Figure 4 nutrients-16-03647-f004:**
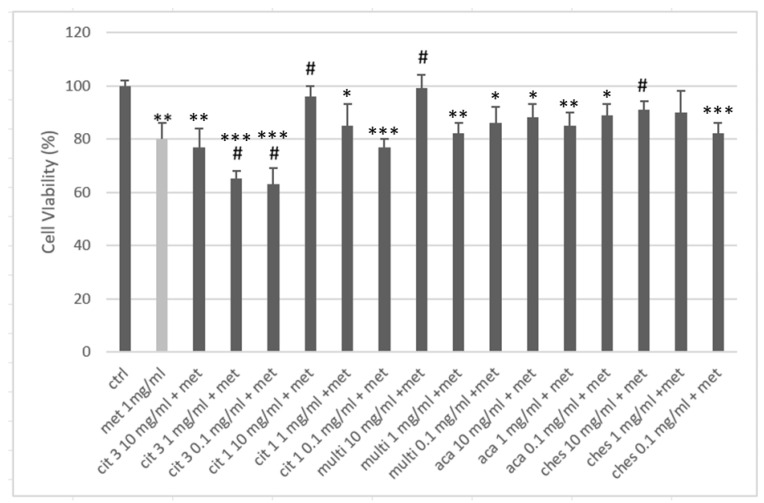
Cell proliferation after 24 h treatment with methylprednisolone (MET) and honey. * = *p*< 0.05 vs. ctrl ** = *p* < 0.01 vs. ctrl *** = *p* < 0.001 vs. ctrl; # = *p* < 0.05 vs. met; (*t*-test).

**Figure 5 nutrients-16-03647-f005:**
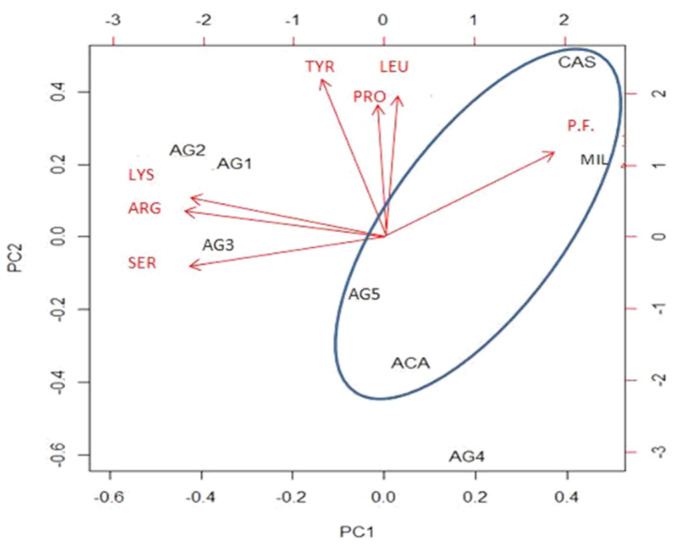
PCA graph with respect to the two components, PC1 and PC2. The red symbols refer to the original variables most closely related to the dimensions PC1 and PC2 (tyrosine—TYR, leucine—LEU, proline—PRO, ornithine—ORN, serine—SER, arginine—ARG, lysine—LYS, polyphenols—P.F.), with arrows indicating direction and intensity of the vector. The black symbols represent the analyzed honeys: citrus honey 1—AG1, citrus honey 2—AG2, citrus honey 3—AG3, citrus honey 4—AG4, citrus honey 5—AG5, chestnut honey—CAS, acacia honey—ACA, multifloral honey−MIL.

**Table 1 nutrients-16-03647-t001:** Water content values of honeys.No statistically significant difference has been revealed by ANOVA followed by Tukey’s post hoc test (*n* = 6).

Honey	%Water Content ± s.d.
Citrus 1	16.40 ± 1.80
Citrus 2	18.00 ± 1.82
Citrus 3	17.10 ± 1.40
Citrus 4	16.40 ± 2.10
Citrus 5	16.20 ± 2.00
Acacia	17.70 ± 2.22
Chestnut	17.50 ± 1.77
Multifloral	17.20 ± 2.45

**Table 2 nutrients-16-03647-t002:** Concentration (mg/mL) ±s.d. of carbohydrates determined in honey samples. Different letters indicate statistically different values (ANOVA followed by post hoc Tukey test, *n* = 6).

Honey	Pectins	Sucrose	Glucose	Fructose	Melezitose
Citrus 1	5.17 ± 1.20 ^a^	78.68 ± 9.82 ^a^	372.10 ± 42.89 ^a^	440.40 ± 22.76 ^a^	4.47 ± 2.20 ^a^
Citrus 2	3.98 ± 1.22 ^a^	73.74 ± 10.43 ^a^	310.10 ± 32.65 ^a^	441.67 ± 30.37 ^a^	3.86 ± 1.43 ^a^
Citrus 3	4.40 ± 2.24 ^a^	81.30 ± 11.76 ^a^	370.15 ± 36.78 ^a^	455.29 ± 21.09 ^a^	4.52 ± 1.89 ^a^
Citrus 4	4.43 ± 1.80 ^a^	90.22 ± 14.45 ^a^	334.33 ± 44.86 ^a^	460.47 ± 19.02 ^a^	5.04 ± 1.92 ^a^
Citrus 5	6.68 ± 2.88 ^a^	84.86 ± 13.65 ^a^	337.29 ± 36.77 ^a^	424.63 ± 26.90 ^a^	4.93 ± 1.64 ^a^
Acacia	5.12 ± 2.76 ^a^	95.75 ± 15.22 ^a^	304.73 ± 44.45 ^a^	532.70 ± 21.64 ^b^	7.18 ± 2.49 ^ab^
Chestnut	16.93 ± 4.20 ^b^	173.59 ± 32.76 ^b^	181.14 ± 32.87 ^b^	374.11 ± 19.33 ^c^	10.45 ± 2.66 ^bc^
Multifloral	14.33 ± 3.80 ^b^	151.66 ± 28.96 ^b^	260.11 ± 42.44 ^b^	421.09 ± 22.97 ^c^	13.30 ± 4.60 ^b^

**Table 3 nutrients-16-03647-t003:** Concentration (nmol/mL) ±s.d. of free amino acids in honey samples. Different letters indicate statistically different values (ANOVA followed by Tukey post hoc test, *n* = 6). Statistics were applied to amino acids common to all the samples.

	Citrus 1	Citrus 2	Citrus 3	Citrus 4	Citrus 5	Acacia	Chestnut	Multifloral
ALA	-	-	546.5 ± 14.7	-	-	-	-	-
β-ALA	1871.9 ± 28.2 ^a^	2506 ± 45.3 ^b^	2105.80 ± 38.9 ^c^	786.47 ± 21.6 ^d^	1516.57 ± 64.8 ^e^	1577 ± 33.7 ^e^	1431 ± 22.8 ^e^	800 ± 19.8 ^d^
ARG	82 ± 10.2	151.2 ± 19.9	49.37 ± 8.5	-	149 ± 16.6	73 ± 9.9	88 ± 10.3	54 ± 7.7
ASP	95.1 ± 11.7	89.55 ± 10.8	85.57 ± 11.7	-	-	128 ± 11.4	164 ± 16.8	131 ± 11.9
GLU	375.5 ± 21.4	545.05 ± 22.7	482.5 ± 18.9	168 ± 16.6	328 ± 19.9	^-^	436 ± 30.2	230 ± 21.3
GLY	148.9 ± 22.4	-	-	-	-	72 ± 11.3	72 ± 9.9	73 ± 10.7
HYS	157.3 ± 28.9 ^ac^	231.7 ± 27.6 ^a^	181.7 ± 19.8 ^a^	80.1 ± 10.6 ^b^	80.5 ± 10.2 ^b^	116 ± 9.8 ^c^	57 ± 8.3 ^b^	57 ± 9.9 ^b^
ILE	32 ± 8.9 ^a^	32.5 ± 7.7 ^a^	28 ± 6.1 ^a^	19.5 ± 7.0 ^a^	19.4 ± 8.1 ^a^	22 ± 7.6 ^a^	17 ± 5.6 ^a^	25 ± 5.7 ^a^
LEU	20 ± 4.7	19.5 ± 5.6	13.3 ± 4.8	-	15.2 ± 3.5	17 ± 6.1	19 ± 5.9	24 ± 7.3
LYS	93.5 ± 13.50 ^a^	118.2 ± 10.6 ^a^	115.7 ± 15.9 ^a^	40.8 ± 8.4 ^b^	66.6 ± 9.3 ^b^	60 ± 9.7 ^b^	60 ± 8.3 ^b^	12 ± 3.1 ^c^
ORN	43.8 ± 10.4 ^a^	24.8 ± 6.7 ^abc^	28.5 ± 8.8 ^abc^	13.6 ± 4.1 ^b^	17.2 ± 3.5 ^b^	13 ± 3.0 ^b^	34 ± 6.6 ^ac^	19 ± 3.2 ^b^
PHE	99.8 ± 11.3	91.0 ± 10.4	85.2 ± 9.7	27.9 ± 4.1	74.7 ± 9.5	-	42 ± 5.7	62 ± 5.3
PRO	3301.6 ± 70.9 ^a^	4429 ± 81.3 ^b^	3725.5 ± 90.2 ^c^	1636.79 ± 61.7 ^d^	2530.1 ± 68.9 ^e^	2743 ± 66.4 ^e^	3515 ± 75.1 ^ac^	4866 ± 82.4 ^b^
SER	606.2 ± 58.9 ^a^	657.2 ± 60.2 ^a^	693.3 ± 68.9 ^a^	304.3 ± 38.6 ^b^	344.5 ± 32.7 ^b^	564 ± 51.3 ^a^	141 ± 17.8 ^c^	187 ± 18.1 ^c^
TAU	65.4 ± 5.3	-	47 ± 3.7	-	-	89.6 ± 7.8	-	21 ± 3.2
THR	86.2 ± 7.9 ^a^	81.5 ± 7.3 ^a^	56.3 ± 6.3 ^b^	19.9 ± 2.3 ^c^	67.6 ± 6.1 ^a^	39 ± 4.2 ^b^	55 ± 4.8 ^b^	43 ± 4.6 ^b^
TYR	56.4 ± 6.2 ^a^	50 ± 5.3 ^a^	49.9 ± 5.1 ^a^	21 ± 2.9 ^b^	31.2 ± 3.1 ^c^	36 ± 3.8 ^c^	64 ± 5.8 ^a^	36 ± 4.1 ^c^
VAL	69.9 ± 7.2 ^a^	79.6 ± 8.2 ^a^	71.2 ± 6.3 ^a^	28.7 ± 3.1 ^b^	43.6 ± 5.1 ^c^	50 ± 4.2 ^c^	72 ± 5.3 ^a^	39 ± 3.9 ^b^

**Table 4 nutrients-16-03647-t004:** Parameters of the UV method employed for the quantification of total flavonoids.

Parameter	Value
R^2^	0.99
Equation	y = 4317.80 − 98.57
Linearity range	0.31–20 g/mL
Limit of quantification	0.3 g/mL
Intra- and inter-day variation	<10%

**Table 5 nutrients-16-03647-t005:** Concentration (mg/kg) ±s.d. of polyphenols (expressed as gallic acid) and total flavonoids (expressed as quercetin) in honeys. Different letters indicate statistically different values (ANOVA followed by post hoc Tukey test, *n* = 6).

	Total Polyphenols (mg/kg)	Total Flavonoids (mg/kg)
Citrus 1	214.60 ± 11.79 ^a^	47.69 ± 1.24 ^a^
Citrus 2	201.92 ± 12.60 ^a^	48.84 ± 1. 71 ^ab^
Citrus 3	208.26 ± 12.42 ^ac^	47.69 ± 2.80 ^a^
Citrus 4	248.42 ± 5.31 ^b^	46.82 ± 1.25 ^a^
Citrus 5	220.94 ± 8.77 ^c^	53.18 ± 3.64 ^b^
Acacia	293.97 ± 11.80 ^d^	58.67 ± 0.92 ^c^
Chestnut	762.13 ± 1.41 ^e^	514.45 ± 2.89 ^d^
Multifloral	688.14 ± 2.33 ^f^	540.46 ± 2.57 ^e^

**Table 6 nutrients-16-03647-t006:** Anti-radical activity, expressed as % inhibition compared to the control, of honeys at a concentration of 10 mg/mL. Different letters indicate statistically different values (ANOVA followed by post hoc Tukey test, *n* = 6).

	%Antioxidant Activity ± s.d
Citrus 1	-
Citrus 2	-
Citrus 3	-
Citrus 4	-
Citrus 5	1.85 ± 0.15 ^a^
Acacia	11.61 ± 1.03 ^b^
Chestnut	38.06 ± 2.34 ^c^
Multifloral	8.77 ± 0.98 ^d^

**Table 7 nutrients-16-03647-t007:** Results(%) of melissopalinological analysis of honeys.

Honey	Chestnut Pollen	Acacia Pollen	Citrus Pollen	Sulla Pollen
Citrus 1	-	-	36	21
Citrus 2	-	-	33	
Citrus 3	-	-	56	
Citrus 4			53	
Citrus 5	-	-	35	22
Acacia	-	25	-	-
Chestnut	92	-	-	-
Multifloral	51	-	-	-

## Data Availability

The original contributions presented in the study are included in the article. Further inquiries can be directed to the corresponding author.

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
