# Peer review of "Eudermic Properties and Chemical–Physical Characterization of Honeys of Different Botanical Origin"

_nutrients, 2024, doi:10.3390/nu16213647_

Round 1
Reviewer 1 Report
Comments and Suggestions for Authors
The manuscript deals with an exciting aspect of using honey, providing an evidence-based approach for exploiting honey in dermatology.
Abstract:
L16: The present work aims to deepen our knowledge / understanding of the activity ... - inserting the text in bold is suggested
L18: the study wants - or: intends
The Introduction provides sufficient background, but I think it is too detailed in certain aspects, e.g. how nectar is collected and processed by bees. In my opinion this is rather general knowledge for researchers who investigate beehive products, thus this part of the introduction should be shortened. The focus should rather be placed on the bioactive compounds of honey and its healing potential, with special emphasis on evidence available so far in dermatology.
Materials and Methods:
L 102: instead of "glass dish", glass lid
2.3. Determination of the Sugar Profile
Table 1. is not necessary, since all the reported parameters are the same when detecting different sugars, except for retention times. So I suggest to include mobile phase, temperature, elution, etc. only in the text, and then highlight the respective retention times for different sugar types.
2.7. Qualitative Melissopalynological Characterization
magnification: was it not 400x or 1000x? In my experience 40x magnification is not enough to observe the unique traits of pollen grains and to determine the pollen type. In several case, 100x magnification is still not enough. So perhaps the magnification mentioned in the text is only the magnification of the objective.
Results:
3.4.
Text below Table 6: no need to repeat the exact numerical data that can be read from the table.
To clarify, what is meant by "In the first" and "In the second", the authors are suggested to write "In chestnut honey" and "In multifloral honey".
3.5. Antioxidant activity
"a clear correlation" - was there an appropriate statistical analysis to claim this relationship?
Can you provide any explanation for the low antioxidant activity of multifloral honey?
3.5.
Table 8 should be extended with percentage values of other pollen types mentioned - these data are partially provided in the text, but it would be much more informative to see these data summarized in the table. The percentage values currently provided in the table should be specified, e.g. 92% is the ratio of Castanea pollen, 25% is the ratio of Robinia pollen, etc.
Figure 1. c) the echinate pollen grain looks rather a goldenrod (Solidago) pollen, not sunflower (Helianthus), please check it
3.6. Collagenase inhibition assay
Again, data should be presented EITHER in text OR in table - I think the text would be enough, there is no really need for using a table here for a single dataset (not a complex data matrix)
L324: efficacy?
The Conclusions are fine, supported by experimental evidence.
In summary, I think that this manuscript presents some novel findings that can be of interest for several researchers and health care professionals. Presenting data should be improved according to suggestions above.
Comments on the Quality of English Language
The English language of the manuscript is generally fine, although some misspellings occur in the text. In addition, in several cases the space is missing between two words, but this is rather a formal mistake, not a language-related one.
Author Response
For my answers, please check the attached file

Reviewer 2 Report
Comments and Suggestions for Authors
The manuscript describes the chemical composition and investigates some biological effects of eight honey samples. It provides a few new data; however, significant improvements are needed.
General comment:
1) The introduction provides relevant background, but the discussion is the weakest point of the manuscript. This section is combined with the results, but is based on only six references. In reality, there is a lack of discussion of the obtained results in the context of available literature. There are numerous studies on the chemical composition and the role of honey in skin repair that should be mentioned and discussed.
The other comments:
2) The abstract should be rewritten. This section should contain the background, aim, methods used, results, and conclusion. In its current version, it only describes the conducted research, lacking, for example, a brief discussion of the results.
3) Introduction: Some references should be added, e.g., for the chemical composition (line 58). The aim of the study should more specifically describe the scope of the investigations.
4) Table 1 is unnecessary because the chromatographic conditions were the same for all compounds. Details about the chromatographic column (manufacturer, length, particle diameter, etc.) should be included in the text. Retention times could be added to the appropriate tables in the Results and Discussion sections. Commonly, when HPLC analysis is included in a manuscript, chromatograms are provided to support the data (these can be added to the main body or as supplementary materials).
5) There is a lack of information about the standards used in the study (e.g. in HPLC analysis). What was their source? Additionally, the manufacturer of the reagents used is not provided.
6) Table 2: "Kit AccQ.Tag" is not a mobile phase; it is simply a reagent used for derivatization. The term "mobile phase" refers to the eluent used in HPLC separation. "Length Wave issue" should be corrected to "emission wavelength."
7) 2.6. "Total Flavonoid Content" – this is not the correct method for determining TFC. TFC is typically based on the reaction with aluminum chloride. It appears that the authors simply measured the absorbance of the samples. Moreover, the results were expressed as hyperoside (line 145) or as quercetin (line 147)? This needs clarification.
8) Line 148: "5000 to 78 mg/L" – 5000 mg/L? This value is extremely high. Additionally, the range should be presented from the lowest to the highest value.
9) All tables lack statistical analysis; they only present mean values. Please add the standard deviation and include statistical analysis to show whether the differences between the values are statistically significant.
10) Table 5 should be placed after its first mention in the text.
11) Line 271: „These data..." – what data do the authors mean? Please be more precise.
12) 3.5. Section: "A clear correlation between polyphenol content and antioxidant activity at a concentration of 10 mg/mL is remarkable" – it is difficult to agree with this statement. For example, the Acacia sample showed 11.61% inhibition and a TPC of 293.97 mg/kg, while the multifloral sample exhibited a significantly higher TPC (688.14 mg/kg) and a lower percentage of inhibition (8.77%).
13) Figures 1 and 2 are not mentioned in the text.
14) Figure 4: "metpred" – the abbreviation should be used consistently (correct to: "met"). Moreover, the statistics are unclear. The symbol "**," which indicates a statistically significant difference versus the control, is only present at met 1 mg/mL. However, for some other values (e.g., cit 3 at 1 mg/mL), the differences appear to be statistically significant. Please also check the placement of the "#" symbol.
15) Figure 5 is unnecessary and could be moved to supplementary material.
16) Figure 6: All abbreviations used in the figure should be explained in the figure legend (please check the other figures as well). Figures should be self-explanatory.
17) The conclusion should be rewritten. This section is too long. The conclusions should provide a brief summary of the most important findings.
18) There are many editorial errors e.g.
Lack of spaces: Line 64: „Honeyhas”, Line 99: „contentmeasurement At thelowerconcentrationstested, allhoneysshowed, etc…. Check the text carefully.
Table 2: eluente?
Line 325: „experimentation”?
Author Response

(The authors gave the same response as above.)

Round 2
Reviewer 1 Report
Comments and Suggestions for Authors
I appreciate the efforts of the authors to improve their manuscript according to my and Reviewer 2's suggestions. The authors did a really good job when revising their MS.
I have two minor comments yet:
Abstract:
L17: please keep EITHER "knowledge" OR "understanding"
Results:
Fig. 1. legend: please use the term "pollen grain" instead of "pollen granule" (Pollen grain refers to the whole dispersal unit of pollen, whereas granule can be a part within the pollen grain, e.g. the allergen-containing cytoplasmic granules released by pollen grains).
Author Response
We sincerely thank the Reviewer
Reviewer 2 Report
Comments and Suggestions for Authors
The manuscript has been improved, but unfortunately, the authors ignored some of my comments. Therefore, I am raising these issues again for the consideration.
1) The discussion is the weakest point of the manuscript. This section is combined with the results and is supported by only six references (eight in the current version). The discussion must be expanded. Numerous studies on the chemical composition of honey and its role in skin repair should be mentioned and thoroughly discussed.
2) The abstract should be rewritten. No results are mention in this section.
3) My comment: 2.6. "Total Flavonoid Content" – this is not the correct method for determining TFC. It appears that the authors simply measured the absorbance of the samples.”
Authors response: The direct reading method was chosen for the quantification of total flavonoids as it has already been validated and used by both our research group (Governa et al., 2019) and others (Sosa et al., 2007) in published work. In addition, the literature shows it to be a more precise method than that with aluminium chloride (Shraim et al., 2021)
I cannot accept the response. As the authors did not provide the full data for the references; thus, I am unable to see the methodologies mentioned in the cited papers (Governa et al., 2019; Sosa et al., 2007). Shraim et al. (2021) provided a critical review of the method using aluminum chloride, but they did not consider other alternatives. Of course, complexation with AlCl₃ has some limitations, but for now, it remains a commonly used method in scientific research for TFC evaluation. Furthermore, while direct absorbance measurement may be more precise, it is certainly less specific. Not only flavonoids absorb at 353 nm—for example, ellagic acid also shows strong absorbance at 350 nm (and many other compounds). The authors simply measured the absorbance of the sample.
4) Tables 2-4. Indicate whether the differences between the values are statistically significant. You can use, for example, different letter symbols to denote values that are significantly different.
5) There are still lack of spaces in many places. Check the text carefully.
Author Response
Please, see attached file. thank you.
